# Charge Modification as a Mechanism for Tunable Properties in Polymer–Surfactant Complexes

**DOI:** 10.3390/polym13162800

**Published:** 2021-08-20

**Authors:** Christopher Hill, Wasiu Abdullahi, Robert Dalgliesh, Martin Crossman, Peter Charles Griffiths

**Affiliations:** 1School of Science, Faculty of Engineering and Science, University of Greenwich, Chatham Maritime, Kent ME4 4TB, UK; christopher.hill@greenwich.ac.uk (C.H.); w.o.abdullahi@greenwich.ac.uk (W.A.); 2ISIS Neutron and Muon Source, STFC, Rutherford Appleton Laboratory, Harwell Oxford, Didcot, Oxford OX11 0QX, UK; robert.dalgliesh@stfc.ac.uk; 3Unilever Research, Port Sunlight, Quarry Road East, Bebington, Wirral CH63 3JW, UK; Martin.Crossman@unilever.com

**Keywords:** polyelectrolyte–surfactant interactions, small-angle neutron scattering, complexation, charge modification

## Abstract

Oppositely charged polymer–surfactant complexes are frequently explored as a function of phase space defined by the charge ratio *Z*, (where *Z* = [${+}_{polymer}]/\left[{-}_{surfactant}\right]$), commonly accessed through the surfactant concentration. Tuning the phase behaviour and related properties of these complexes is an important tool for optimising commercial formulations; hence, understanding the relationship between *Z* and bulk properties is pertinent. Here, within a homologous series of cationic hydroxyethyl cellulose (cat-HEC) polymers with minor perturbations in the degree of side chain charge modification, phase space is instead explored through $\left[{+}_{polymer}\right]$ at fixed C${}_{polymer}$. The nanostructures were characterised by small-angle neutron scattering (SANS) in D${}_{2}$O solutions and in combination with the oppositely charged surfactant sodium dodecylsulfate (h- or d-SDS). Scattering consistent with thin rods with an average radius of ∼7.7 Å and length of ∼85 Å was observed for all cat-HEC polymers and no significant interactions were shown between the neutral HEC polymer and SDS (C${}_{SDS}$ < CMC). For the charge-modified polymers, interactions with SDS were evident and the radius of the formed complexes grew up to ∼15 Å with increasing *Z*. This study demonstrates a novel approach in which the *Z* phase space of oppositely charged polymer–surfactant complexes can be controlled at fixed concentrations.

## 1. Introduction

Many industrial processes rely on controlling the tunable behaviour of formulated products through an appropriate blending of oppositely charged polymer–surfactant mixtures. It is well known that these mixtures interact in a strongly synergistic way and selective control of the phase behaviour can be achieved through adjustment of the charge ratio *Z*, (where *Z* = [${+}_{polymer}]/\left[{-}_{surfactant}\right]$) via the relative concentrations of the polymer and surfactant charges. As a result of this relative ease of control over such properties, oppositely charged polymer–surfactant mixtures have had great success as formulated products in applications such as, but not limited to, detergency [1,2,3], drug delivery [4,5,6,7], and rheological modifiers [8,9].

Indeed many of the applications that employ oppositely charged polymer–surfactant systems make use of synthetic polymers such as anionic poly(acrylic acid) or cationic poly(diallyldimethylammonium chloride) [2,5,10,11,12,13]. However, more recently, polysaccharides have emerged as an important component in commercially formulated products as a result of their biocompatibility, biodegradability, bioadhesivity, and nontoxicity [14,15,16,17,18,19,20,21,22,23,24]. One series of polysaccharide-based polyelectrolytes (PEs) that have received particular attention due to their commercial relevance and interesting rheological properties are cationic hydroxyethyl cellulose (cat-HEC) polymers, in particular, JR400 [14,15,16,17,18,20,23,25]. In oppositely charged PE–surfactant complexes, around charge neutrality (where *Z* = 1), it is common to observe an extended region of precipitate formation, while in the region of excess surfactant (*Z* < 1) or PE (*Z* > 1), soluble complexes are formed [2,19]. As a result of this diverse aggregation behaviour, there has been much interest in studying the complex structures formed by these cat-HEC PEs with oppositely charged surfactants in the semidilute [15,16,17,18] and concentrated regime [25].

Through in-depth small-angle neutron scattering (SANS), dynamic light scattering (DLS), and rheological measurements, comprehensive physiochemical characterisation of mixtures containing JR400 and small-molecule surfactants have been carried out [15,16,17,18,20,23,25]. At sufficiently high PE concentrations (>0.5 wt%), addition of low concentrations of small-molecule surfactants sodium dodecylsulfate (SDS) or sodium dodecylbenzenesulfonate (SDBS) produced viscosity increases up to four orders of magnitude higher than that of the pure PE [15,17,18,24]. This viscosity increase was highest in systems with an excess of PE charges, i.e., just before phase separation. In the region of excess surfactant charges (*Z* < 1), however, the precipitate becomes resolubilised by the surfactants and the viscosity drops to below that of pure PE.

Hoffmann et al. have carried out several SANS studies to elucidate the structures formed by these JR400/small-molecule-surfactant mixtures [15,17,18,25]. Early investigations [17] showed that at *Z* values > 2, the viscosity enhancements are explained by the presence of networks of interconnecting rodlike surfactant–PE complexes that increase in length (*L* = 600–800 nm) with increasing surfactant content. High-intensity low-Q scattering suggested that many PE chains were present at this composition, accounting for the interconnection of the PE rods [17]. As the concentration of surfactant is increased past the phase boundary (*Z*≤ 0.1), spherical aggregates were observed with a size commensurate with the pure surfactant micelles, assumed to be in a pearl necklace formation along the PE chain [25]. Contrast variation SANS experiments provided further insight into the structures of these mixtures, with results showing that the PE is preferentially located on the outside of the rodlike aggregates, but is able to penetrate deeply into the surfactant-dominated core.

Investigations on a series of cat-HEC PEs with minor perturbations in the degree of side chain charge modification were characterised by both pulsed-gradient spin-echo NMR (PGSE-NMR) and electrophoretic NMR (eNMR) with and without the presence of SDS [14]. PGSE-NMR results highlighted substantial reductions in PE diffusion by addition of SDS to the charged systems, arising due to surfactant-bridging interpolymer entanglements, mirrored by increases in viscosity. Irrespective of the degree of polymer modification, a linear relationship between the charge on the PE–surfactant complex and the level of surfactant binding was observed by eNMR measurements. Interestingly, the PE with the highest degree of charge modification (*N* = 2.7%) displayed lower levels of surfactant binding, which was hypothesised as being a result of competing counterion condensation effects such that a smaller number of surfactant molecules are accommodated on the polymer.

Here, we build upon previous findings [14] to fully understand how a range of cat-HEC PEs with differing degrees of charge modification interact with small-molecule surfactant SDS. This is carried out by exploiting the ability of altering *Z* via both $\left[{+}_{PE}\right]$ and $\left[{-}_{SDS}\right]$; hence, scanning a range of *Z* values is possible at both fixed C${}_{polymer}$ and C${}_{SDS}$. This represents an interesting lacuna within the scientific literature, which would provide a novel approach to controlling the phase behaviour and related bulk properties of oppositely charge polymer–surfactant complexes. Taking advantage of the unique ability of SANS to vary the contrast of a sample by varying the isotopic composition, this study aims to highlight regions of interest within the aggregates to gain an in-depth understanding of how the PE and surfactant interact within these complex systems. Furthermore, this will lead to improved understanding into the structure–property relationship and associated dynamics in such complexes and demonstrate the ability of finely adjusting the characteristics of formulated products at fixed concentrations as required.

## 2. Materials and Methods

### 2.1. Materials

Sodium dodecylsulfate (SDS) (Sigma Aldrich, Gillingham, UK, ≥99.0%) and deuterium oxide (D${}_{2}$O) (Sigma Aldrich, Gillingham, UK, ≥99.9%) were used as received. Deuterated sodium dodecylsulfate was synthesised by the ISIS deuteration facility and has been used as received. Quaternised hydroxyethyl cellulose polymers have a cellulose backbone and may be regarded as polymeric, quaternary ammonium salts of hydroxyethyl cellulose that have been reacted with trimethylammonium-substituted epoxide (structure shown in Figure 1). Five different quaternised hydroxyethyl cellulose polymers, kindly supplied by Dow Chemical Company, with different degrees of modification denoted *N* (cationic substitution), have been employed here. Cationic substitution refers to the amount of trimethylammonium substitution along the polymer backbone and, in this study, is noted by the nominal degree of modification, expressed in terms of the percentage nitrogen content: *N* = 0%, *N* = 0.5 (±0.1)%, *N* = 0.95 (±0.15)%, *N* = 1.8 (±0.2)%, *N* = 2.7 (±0.2)%.

### 2.2. Methods

#### Small-Angle Neutron Scattering (SANS)

SANS measurements were performed on Larmor at the ISIS facility (Rutherford Appleton Laboratory, Didcot, UK). On Larmor, a simultaneous *Q*-range of 0.003–0.7 Å${}^{-1}$ was achieved with a neutron wavelength range of 0.9 < λ < 13.5 Å. All samples were made in D${}_{2}$O, using 2-mm path length rectangular quartz cells at a temperature of 25 ${}^{\circ }$C. Raw SANS data were reduced by subtracting the scattering of the empty cell and the D${}_{2}$O background and normalised to an appropriate standard using the instrument-specific software. SANS data were fit using the analysis package SasView.

In a SANS experiment, the intensity (*I(Q)*) of scattered neutrons is measured as a function of momentum transfer (*Q*):(1)$$Q=\frac{4\pi sin\theta }{\lambda }$$

For a monodispersed homogeneous scattering system of radius *R* in a solvent, the normalised SANS intensity *I(Q)* (cm${}^{-1}$) is as follows:(2)$$I\left(Q\right)=\left(\frac{N}{V}\right)V_{p}^{2}\Delta \rho^{2}P\left(Q,R\right)S\left(Q\right)$$
where ($\frac{N}{V}$) is the number density of particles, $V_{p}$ is the particle volume, and $\Delta \rho^{2}$ is the difference in SLD between the scatterer $\rho_{\mathrm{SLD}}$ and the solvent $\rho_{s}$. The first three terms in Equation (Equation 2) are independent of *Q* and account for the absolute intensity of scattering. This is often referred to as the scale factor, $S_{F}$, which can be defined as
(3)$$S_{F}=\left(\frac{N}{V}\right)\Delta \rho^{2}V_{P}^{2}=\phi_{p}\Delta \rho^{2}V_{P}$$
where $\phi_{p}$ is the volume fraction of the particles. The scale factor provides a measure of the validity of a model when analysing SANS data, i.e., the $S_{F}$ determined from the fit can be compared to the calculated value. The last two terms in Equation (Equation 2) are *Q*-dependent functions. *P(Q, R)* is the particle form factor, which describes intraparticle information such as size and shape. *S(Q)* is the structure factor, which describes the scattering due to interparticle correlations.

## 3. Results and Discussion

In the first instance, the nanostructures formed by the series of cat-HEC PEs at a practical concentration of 1 wt% are investigated individually before discussions of mixed cat-HEC/SDS complexes. This is so that the polymer-only scattering can be fully resolved and compared with the relevant literature prior to discussions regarding the cat-HEC/SDS complexes.

### 3.1. Polymer-Only Solutions

Figure 2 presents the SANS profiles for the family of cat-HEC PEs in aqueous (D${}_{2}$O) solutions at a fixed concentration of 1 wt%. As has been previously reported for similar PEs in the semidilute regime, weak scattering is observed for all cat-HEC polymers in this study [15,16,17,18,25], being consistent with the presence of thin, rodlike solution structures. Despite this similarity in scattering intensity, some differences are observed in the shape of the scattering profiles as the degree of side chain modification is increased.

Hoffmann et al. reported in several studies the structural information of a 1 wt% JR400 PE with cationic groups on 27% of the glucose units, equating to charge modification of ∼1.5% [15,17,18,25]. In these reports, the scattering data for JR400 were best modelled with a compound model that appropriately describes the different regions within the scattering data. The model contained a form factor of a thin rod to describe the data within mid to high Q range (0.01–0.5 Å${}^{-1}$) and a fractal dimension to represent the low Q region. The overall consensus of these reports, as well as more recent data [16], describe JR400 as thin rods with a length of ∼65 Å, radius of ∼8 Å and a low Q fractal dimension of 3.

Prior to discussions of the fitted structural parameters of the cat-HEC polymers, it is instructive to initially discuss the gross features of the data, considering first the uncharged polymer. In the low-Q region, scattering scales as I(Q) ≈ Q${}^{D}$, where D is a characteristic dimensionality of the dispersed PE and the gradient associated with a log–log plot will be -D [26]. For commonly observed form factors such as spheres, cylinders, and disks, D values will be 0, 1, and 2, respectively. In the absence of any side chain modification (*N* = 0%), the observed scattering follows a Q${}^{-1}$ dependence throughout the measured Q range, being consistent with the formation of thin, rigid, rodlike structures [26].

We now consider the charged polymers. In the high Q region, a Q${}^{-1}$ dependence is present in all PEs, indicating the short-range polymer conformation. However, as the degree of charge modification is increased, there is a levelling off of the scattering intensity and a transition towards a Q${}^{0}$ within the mid Q region. This behaviour is more discernible in PEs that contain lower levels of charge modification, i.e., *N* = 0.5% and *N* = 0.95%, but is obfuscated at higher levels of charge modification due to the emergence of a repulsive structure factor (S(Q)). Furthermore, all PEs with side chain modifications show a large increase in intensity with a Q${}^{-3}$ dependence within the low Q region (Q < 0.02 Å${}^{-1}$), indicating the formation of large PE networks with domain structures bearing sharp interfaces [27,28]. This provides evidence that these large network structures are formed by repulsions between the charged units on the cat-HEC polymers, leading to expansions in the size of polymer structure. Full resolution of the size/shape of the polymers is therefore not possible within this Q range; instead, the scattering is sensitive to other facets of the local conformation such as the size/shape of the rodlike units that make up the larger structure. The hydrodynamic radius of these polymers is typically 25 nm [14], i.e., the length scale that is beyond the Q range.

As discussed above, accurately modelling these data requires a compound model that appropriately describes the data within the full Q range [15]. In the mid to high Q region, the data were fit using a form factor for a cylinder and the low Q data were fit with a fractal dimension, i.e., the *P(Q)* term in Equation (Equation 2) becomes
(4)$$P\left(Q\right)=aP{\left(Q\right)}^{cylinder}+bP{\left(Q\right)}^{fractal}$$
where *a* and *b* are the amplitudes of the cylinder and fractal form factors, respectively. The cylinder term accounts for the scattering arising from the cat-HEC PE structure (and PE–SDS complexes below) and the fractal term accounts for the assembly of PE network structure.

The parameters obtained from modelling of the cat-HEC PEs at 1 wt% are reported in Table 1, and additional modelling parameters are in the Appendix A. Overall, the compound model reproduced the data well for all cat-HEC polymers ($\chi^{2}$≤ 3), successfully describing the emergence of a subtle mid-Q peak as the degree of charge modification is increased. Low Q data on the other hand were found to be slightly overrepresented with this model; however, this did not adversely influence the fitted structural parameters, which remained consistent with the literature [15,16,18,25].

The data clearly show that side chain modification has an effect on the local conformation of the PE, as seen by the increase in radius (*R*) and decrease in length (*L*) as the degree of modification is increased. When considering the fractal dimensions within the low Q region, values of 1, 2, or 3 represent line, ideal, or compact chain structures, respectively [29]. All modified PEs in this study have a fractal dimension of ∼3 and, therefore, are best represented as compact network structures within the low Q region. The overall increase in *R* with increasing charge modification is likely to be caused by charge repulsion between the PE units. As more side chain charge modification is introduced into the PE, the charged moieties will repel each other and the size of the rodlike units will expand. Interestingly, the length of the unmodified PE units is ∼3 times longer than the modified PEs. The reason for this has been confirmed as being a result of the chemical modification process that has an effect on the *L* of the PE units.

### 3.2. Cat-HEC/SDS Complexes in the Semidilute Regime

Having characterised the aggregation behaviour of the 1 wt% PEs individually, any structural changes induced by the addition of the surfactant will be evident when considering the scattering patterns of the mixed PE–SDS complexes. When d-SDS is employed, no significant scattering from the surfactant is evident, so the changes in scattering pattern arise due to the changes in the polymer concentration. In all of these measurements, the concentration of PE is fixed at 1 wt% and SDS is present at either 2 or 4 mM, allowing for *Z* to be explored via both $\left[{+}_{PE}\right]$ and $\left[{-}_{SDS}\right]$. To resolve the full structure of the aggregates, two contrasts have been employed in this study: a full contrast (cat-HEC, h-SDS, D${}_{2}$O) and a PE contrast (cat-HEC, d-SDS, D${}_{2}$O), which, through their differences, provide an insight into the distribution and aggregation of the surfactant.

#### 3.2.1. Unmodified HEC–SDS Complexes

Consider first the data for SDS and HEC polymer with 0% charge modification in Figure 3. It is clear that there are many similarities when comparing these data; however, some subtle features must be considered. The Q${}^{-1}$ feature from the pure cat-HEC polymer still remains in the SDS/cat-HEC complexes (Q${}^{<-1}$ for 2 mM SDS), suggesting that thin, rodlike polymer structures remain dominant in solution. Additionally, the significant overlap observed between the different data sets demonstrates that addition of SDS does not affect the size or structure of the polymer rods and that the concentration of material being probed remains similar. This is further evidenced by both the fitted structural parameters shown below in Table 2 and the similarities between the scattering of the full and surfactant contrasts (Figure 3). The fitted structural parameters show that within associated errors, the size of the complexes does not change significantly compared to the pure polymer.

One subtle difference between the data in Figure 3 is the intensity increase within the low Q region and change in Q dependency from −1 to ∼−3 associated with the HEC/2-mM SDS complexes. Similar to the PEs bearing charge modification, the fitted fractal dimension within the low Q region is 3, i.e., suggesting formation of mass fractal polymer networks [29]. These same gross features are observed in the surfactant scattering contrast (Figure 3b); however, no other structural changes are seen. Increasing the concentration of SDS (HEC + 4 mM SDS) does not lead to these same features being observed, suggesting that the HEC + 2 mM SDS data are possibly anomalous. PGSE NMR has been used as an additional tool to investigate what influence, if any, does the addition of SDS have on the diffusion and, hence, size of the HEC polymer. These data are reported in the Appendix A and results show no significant change in the diffusion coefficients of either the cat-HEC polymer or SDS when mixed, indicating a lack of significant interaction between the two components. Patel et al. previously reported these same observations using PGSE NMR and, hence, provide additional validity to the argument [14].

This assertion is further justified by comparing the two studied contrasts. No noticeable differences are observed between the full/PE contrasts in either of the cat-HEC/SDS mixtures. If there was significant interaction between the SDS and HEC polymer, it would be expected that some slight changes in intensity throughout the Q-range would be seen to reflect changes in the scattering length density or volume fraction of the scatterers (i.e., Equation (Equation 2)).

Unlike mixtures containing oppositely charged PEs–surfactants whereby the dominant interactions are between the charges on the respective components, interactions between ionic surfactants and nonionic cellulose polymers occur between the hydrophobic tail of the surfactant and either the polymer backbone or hydrophobic substituent groups [30,31]. Increasing the level of hydrophobic modification within the polymer increases the level of interaction between ionic surfactants and nonionic cellulose polymers [30,31]. Previous literature investigating HEC–SDS mixtures in dilute conditions show that only relatively weak interactions occur between the polymer and surfactant molecules as a result of the low level of hydrophobic modification [14,31,32]. Results reported here are therefore in keeping with previous literature and indicate that any transient hydrophobic interactions are not significant between this series of HEC polymers and SDS.

#### 3.2.2. Charge Modified Cat-HEC/SDS Complexes (Full Contrast)

Oppositely charged PE–surfactant mixtures are known to interact strongly in both the dilute and semidilute regime and, as such, show very interesting aggregation behaviour, which can be fully resolved through the use of contrast variation SANS [2,10,15,18,19,21,33,34]. Here, the nanostructures formed when low concentrations of SDS (C${}_{SDS}$ < CMC) are mixed with cat-HEC polymers bearing differing degrees of charge modification are investigated as a way of exploring and controlling *Z* phase space at fixed C${}_{polymer}$. The SANS profiles, along with fitted I(Q), for the family of cat-HEC PEs mixed with 2 or 4 mM h-SDS (full contrast) are presented below in Figure 4.

Initial observations of these data provide clear evidence of the interaction between the cat-HEC PEs and SDS when compared with unmodified HEC–SDS mixtures in Figure 3. Considering the gross features within the data sets, it can be seen that addition of SDS to the cat-HEC PE leads to an increase in intensity within the mid-Q region with a ∼Q${}^{-1}$ dependency, showing formation of cylindrical shapes aggregates. Compared to the cat-HEC PEs individually, both the radius and length of the complexes are enhanced with addition of SDS, as indicated by both the steeper upturn in the data between ∼0.1 and 0.2 Å${}^{-1}$ and elongation of the ∼Q${}^{-1}$ region. Within the low Q region (Q < 0.01 Å) of some complexes, most prominently in PE = 0.5/0.95% + 4 mM SDS, a sharp increase in intensity is observed with a Q${}^{-4}$ dependency representing formation of compact network structures originating from the PE network as *Z* tends towards 1, i.e., charge neutralisation. As with the cat-HEC polymers individually, this indicates that full resolution of the size/shape of the complexes is not possible within this Q range and instead the scattering is sensitive to local changes in size/shape of the units.

To remain in line with the objectives of this study, it is important that the data are discussed and explored in the context of calculated *Z* values, as indicated in Figure 4. It should be noted that most samples are below charge neutralisation (*Z* > 1) and have an excess of PE charges, i.e., [${+}_{polymer}$] > [${-}_{surfactant}$]. From this perspective, it is clear that the samples that show formation of the compact polymer network (Q${}^{-4}$ dependency in low Q region) are those with low *Z* values (<1.5), i.e., close to charge neutralisation and macroscopic phase separation [15,35,36]. These data therefore indicate the possibility of controlling the bulk properties of these systems at both fixed concentrations through modulation of the [${+}_{polymer}$] or altering the [SDS]. Interestingly, there are significant intensity increases within the low Q region in the samples that contain the cat-HEC PE with the highest degree of charge modification (*N* = 2.7%) in Figure 4d. The high charge density of this cat-HEC polymer accounts for these observations (i.e., [${+}_{polymer}$] is much greater than [${-}_{surfactant}$]) and competitive counterion condensation effects will begin to influence the behaviour of the complexes [37].

For samples with low *Z* values, the emergence of weak interference peaks can be seen, representing a repeating unit comprised of bound aggregated surfactant molecules. The emergence of these interference peaks are masked in the PE contrast samples, indicating that they occur as a result of the surfactant. The spacing between these repeating units (d) can be estimated using Q${}_{max}$ and d = 2π/Q${}_{max}$, as shown in Figure 5. It was not possible to obtain Q${}_{max}$ peaks for samples with *N* = 2.7%; hence, Q${}_{max}$ peaks were averaged from both the 2 and 4 mM SDS samples (*N* = 0.5, 0.95, 1.8%) and modelling allowed for a prediction for *N* = 2.7%. The molecular weight of the family of PEs are expected to remain consistent and, hence, with increasing degree of modification, the charges along the PE backbone will become closer together; Figure 5 therefore displays the reduction in spacing between the surfactant aggregates with increasing degree of charge modification.

The complexes in Figure 4c both show a levelling off in the scattering data towards the low Q limit and, by directly comparing the I(0) values of these two samples, it will be possible to gain quantitative insights into the concentration of material being probed. The I(0) values are ∼0.4 and 2.5 for the 2 and 4 mM-containing SDS samples, respectively. This factor of ∼6 increase in intensity is therefore inconsistent with a doubling of the SDS concentration and suggests that increasing the concentration of SDS in these systems leads to further incorporation of the cat-HEC PE into the nanostructures. To explore this further, the fitted structural parameters of these complexes must be considered.

There have been several studies that have utilised contrast variation SANS to fully elucidate the aggregates formed by oppositely charged cat-HEC polymer and small molecule surfactants, most of which have focused on the cat-HEC polymer JR400 (similar to *N* = 1.8% in this work) [15,16,17,18,20,23]. A mutual conclusion from these investigations have shown that within the PE excess regime (Z > 1), mixed rodlike aggregates are produced with a significantly larger size than the constituent PE. These rodlike surfactant–PE aggregates are responsible for the measured high degree of interconnection between PE chains and resulting significant increase in solution viscosity near the phase boundary on the PE-rich side of the phase diagram [14,15,16,17]. Here, the models are sensitive to local changes in the conformations of the complexes and will focus on how both the degree of charge modification and [h/d-SDS] can be used to influence the size, shape, and related bulk properties of the studied complexes. It is important to reiterate that the models used here are consistent with the literature [15,16,17,18]. The change in radius of the complexes (full and PE contrast) with increasing degree of charge modification are shown in Figure 6, the fitted structural parameters are reported in Table 3.

Figure 6 demonstrates a clear increase in the radius of the cat-HEC/SDS complexes as the degree of polymer charge modification is modulated. The steepest increase in radius is observed between 0–1% degree of modification; these mixtures are close to charge neutralisation and no significant repulsion between the positive charges along the polymer backbone are expected. For cat-HEC PEs with a degree of modification above 1%, the radius of the complexes remains relatively stable around 15 Å; these complexes have an overall excess in the [${+}_{polymer}$] and hence, larger *Z* values. Larger *Z* values indicate that the PE in these complexes will have an excess of positively charged groups along the PE backbone, which can cause additional repulsion and growth within the PE structure. This charge-repulsion-driven growth behaviour is also displayed in the polymer-only data set in Figure 6, whereby the radius of the cat-HEC polymer units increases with degree of modification. It is interesting to note the significant influence that changing the degree of charge modification has on the size of the complexes when compared with altering the [SDS], further indicating the possibility of gaining additional control over the size and related bulk properties of these complexes at fixed concentrations by adjustments to the degree of charge modification.

Several of the studied complexes have similar or comparable values of *Z* (see Table 3); however, they display slight differences in structural parameters. Consider, for example, cat-HEC PE *N* = 0.95% + 2 mM and cat-HEC PE *N* = 1.8% + 4 mM. In this case, *Z* is kept at a constant value of 3, but the fitted *R* value increases as both the number of charges and density of charges along the PE backbone increases across the systems (Figure 6). These slight differences may pertain to the distribution of charges along the PE as well as competitive counterion condensation effects as a result of an increase in dynamically associated micelles that are weakly attracted.

To gain quantitative insights into how SDS is interacting and binding to the different cat-HEC polymers, it is instructive to calculate the number of polymer-bound surfactants in these complexes or aggregation number ($N_{\mathrm{agg}}$) using the following equation [17]:(5)$$N_{\mathrm{agg}}=\frac{V_{\mathrm{agg}}}{V_{\mathrm{surf}}}\cdot \frac{\phi_{\mathrm{surf}}}{\phi_{\mathrm{fit}}}$$
where $V_{\mathrm{agg}}$ is the volume of the polymer–surfactant aggregate, calculated from SANS modelling parameters, $V_{\mathrm{surf}}$ is the calculated surfactant molecular volume, $\phi_{\mathrm{surf}}$ is the calculated volume fraction of surfactant, and $\phi_{\mathrm{fit}}$ is the fitted volume fraction of the polymer–surfactant aggregate from SANS modelling. Calculated $N_{\mathrm{agg}}$ values for the are shown in Table 3, no values for *N* = 2.7% could be calculated for the reasons discussed below. The relatively large errors associated with the calculations are a result of cumulative error associated with the parameters used in Equation (Equation 5). It is interesting to note that the calculated $N_{\mathrm{agg}}$ values are all below that of an ordinary SDS micelle, i.e., $N_{\mathrm{agg}}$ < 60 [38,39,40].

#### 3.2.3. Charge Modified Cat-HEC/SDS Complexes (PE Contrast)

The PE contrast samples (unfilled squares and down triangles in Figure 6 and scattering profiles in Figure 7) allow for an understanding of whether the structure of the cat-HEC PE is changed with addition of SDS. It is important to consider these samples in the context of calculated *Z* values to determine the level of available positively charged groups along the PE backbone. Similar to the full contrast, formation of compact polymer networks (Q${}^{-4}$ dependency in low Q region) are displayed in samples that are close to macroscopic phase separation (*Z* < 1.5), a behaviour that is tunable by increasing the degree of charge modification at fixed concentrations. For most of the samples, the PE contrast shows that thin rods remain as the preferred solution structure for the cat-HEC polymers, though with a radius smaller than those of the parent cat-HEC PEs. This shows that the [${-}_{surfactant}$] sufficiently neutralise the positively charged groups within the PE backbone, leading to less repulsions between the PE units and formation of a more compact structure [41,42,43]. Further evidence for this can be seen in Figure 7, whereby the 0.5, 0.95, and 1.8% degree of modification cat-HEC-containing complexes (PE contrast) display an apparent loss of repulsive S(Q) peaks when compared with the PE only samples, emphasising the loss of repulsions between the PE units.

Complexes containing SDS and cat-HEC PE with *N* = 2.7%, shown in Figure 7d, do not display behaviour that is entirely consistent with the other studied PE systems. Generally, the PE contrast samples have smaller fitted *R* values when compared with full contrast and display scattering similar to that of the PE-only scattering, but with observed suppression of the S(Q) repulsion. Firstly, considering the mixture containing cat-HEC PE with *N* = 2.7% and 2 mM SDS, the recorded scattering profiles are highly comparable in both studied contrasts, reflected also in the similar fitted values of *R* in Table 3. Similarities between these two studied contrasts suggest that there is a relative reduction in the binding of SDS to this high charge density cat-HEC PE as a result of competitive counterion condensation effects, such that fewer surfactant molecules are able to interact successfully with the charged groups on the PE. For high-density PEs, the formation of complexes in which most of the counterions are excluded is expected and commonly reported [14,37].

Consider now the data for the mixture containing cat-HEC PE with *N* = 2.7% and 4 mM SDS. For the 2/4 mM samples in Figure 4d, i.e., the full contrast samples, the overall shapes of the scattering profiles remain grossly similar, with the only notable difference occurring within the mid Q region, whereby the emergence of a weak S(Q) is observed in the 2 mM SDS containing sample. The emergence of this weak S(Q) feature occurs as a result of repulsions within the PE chain as discussed above and seen in the PE-only scattering, and loss of this S(Q) therefore reflects a reduction in repulsive interactions within the PE. Loss of the weak S(Q) occurs as the [${-}_{surfactant}$] is increased (*Z* reduced) and, hence, excess PE charges are able to be neutralised. Interestingly, when comparing the full and PE contrast for the *N* = 2.7% and 4 mM SDS mixtures, a change in Q dependency within the mid Q region is observed. In the full contrast, the mid Q region displays an extended Q${}^{-2}$ dependency; this is reduced to Q${}^{-1}$ in the PE contrast, consistent with scattering from planar surfaces and thin rods, respectively [26,29].

This observed difference between the two studied contrasts suggests that there are two different PE-containing structures in the solution. The PE contrast displays scattering consistent with the *Z* = 6 data for the *N* = 1.8% PE sample in Figure 7c, with similar fitted radii being reported (see Table 3). As discussed above, competing effects such as counterion condensation can reduce binding of surfactants in oppositely charged PE–surfactant complexes [13,14,37], and this may explain the observed change in both Q dependency (Q${}^{-2}$) and structure of the PE when considering the full contrast. The outlying behaviour of the *N* = 2.7% cat-HEC complexes compared with the other samples demonstrates the impact that competitive effects can have on the interactions between an oppositely charged polymer and surfactants. From this study, it is clear that control over the phase behaviour is possible through modulating the charge on a series of polymers; however, it is important to be aware of these competing effects in polymers that contain a high charge density.

## 4. Conclusions

In this work, a homologous series of cat-HEC polymers with different degrees of charge modification were used to explore charge ratio (*Z*) phase space at fixed bulk concentrations through adjustments of the polymer–surfactant charge concentrations ([${+}_{polymer}$] or [${-}_{surf}$]). Contrast variation SANS was used to fully elucidate the nanostructures formed by the family of cat-HEC polymers and SDS as a function of *Z* phase space and demonstrated the ability of controlling the phase properties of the mixtures at fixed concentrations through adjustments of the polymer charge modification.

Individually, the family of cat-HEC PEs displayed very weak Q${}^{-1}$ scattering consistent with thin rods at a practical concentration of 1 wt% as well as intense Q${}^{-3}$ scattering, showing the presence of large polymer network structures. As the degree of charge modification was increased throughout the series, the scattering intensity levelled off towards Q${}^{0}$, reflecting a reduction in the length of the polymer, and for PEs, with *N* > 0.95%, a repulsive S(Q) emerged, indicating repulsions between the polymer units. A relationship between the radius (*R*) of the nanostructures formed by the PEs and the degree of charge modification was shown, with the size of the PE increasing with degree of modification as a result of increased repulsion between the PE units. The modelled structural parameters show that, on average, the charge modified PEs have a *R* of ∼7.7 Å and length of ∼85 Å.

With the addition of SDS (C${}_{SDS}$ < CMC), the unmodified HEC polymer (*N* = 0%) displayed no change in either the shape or intensity of the scattering profiles, both of which would have been indicative of interactions between the two components. Any possible hydrophobic interactions are therefore not significant between this family of cat-HEC polymers and SDS at the studied concentrations. Clear interactions emerged with the introduction of charge modification into the cat-HEC polymer, with the radius of the complexes growing as the degree of modification was increased up to *N* = 1.8%. For the cat-HEC PE with the highest level of charge modification (*N* = 2.7%), the size of the complexes plateaued as a result of the overall excess of $\left[{+}_{polymer}\right]$ and the interactions between the PE and SDS were highly suppressed as a result of competitive counterion condensation effects. For cat-HEC PEs with lower levels of charge modification (*N* = 0.5/0.95%), increasing the concentration of SDS did not lead to significant changes in the size of the complexes; instead, increased scattering in the low Q region was observed. This shows the formation of large polymer network structures as *Z* tends towards 1 (charge neutralisation), behaviour that is commonly mirrored by significant increases in solution viscosity. Use of both h and d-SDS allowed for different regions of the nanostructures (Full and PE contrast, respectively) to be highlighted. Thin rod structures remained as the preferred cat-HEC PEs solution structure in the complexes, but the fitted radii were reduced when compared with the parent cat-HEC PE as a result of charge neutralisation (loss of S(Q)).

Overall, this work has demonstrated a new way of exploring and adjusting the *Z* phase space at constant bulk concentration through modulating the polyelectrolyte–surfactant charge concentrations, allowing for further control over the tunable properties of formulated products containing oppositely charged polymer–surfactant mixtures.

## Figures and Tables

**Figure 1 polymers-13-02800-f001:**
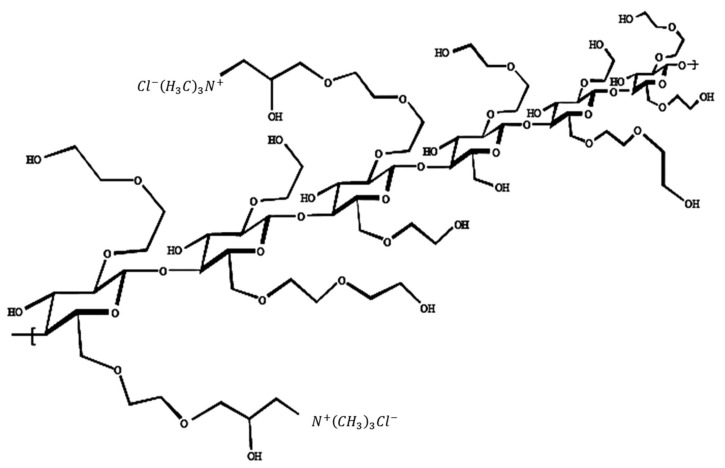
Generic structure of cationic hydroxyethyl cellulose (cat-HEC) polymers.

**Figure 2 polymers-13-02800-f002:**
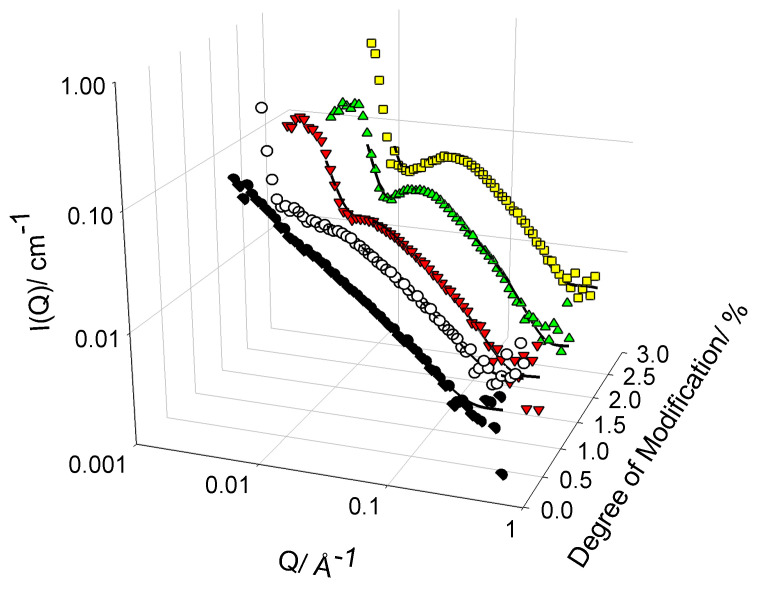
Small-angle neutron scattering (SANS) profiles from 1 wt% polymers with different degrees of charge modification.

**Figure 3 polymers-13-02800-f003:**
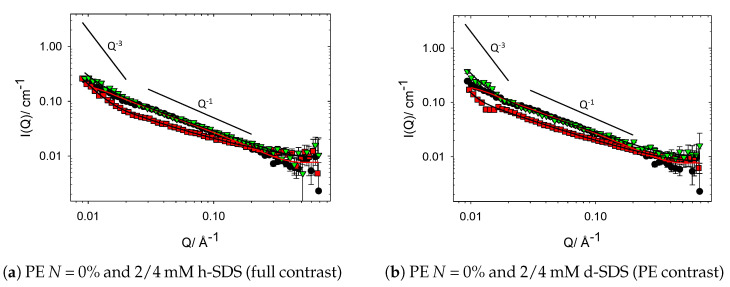
SANS profiles from binary mixtures containing 1 wt% HEC polymer and h-SDS (full
contrast) or d-SDS (PE contrast). Symbols: circles, no SDS; squares, 2 mM SDS; down triangles, 4 mM
SDS. The solid lines show Q^−1^ and Q^−3^ behaviour.

**Figure 4 polymers-13-02800-f004:**
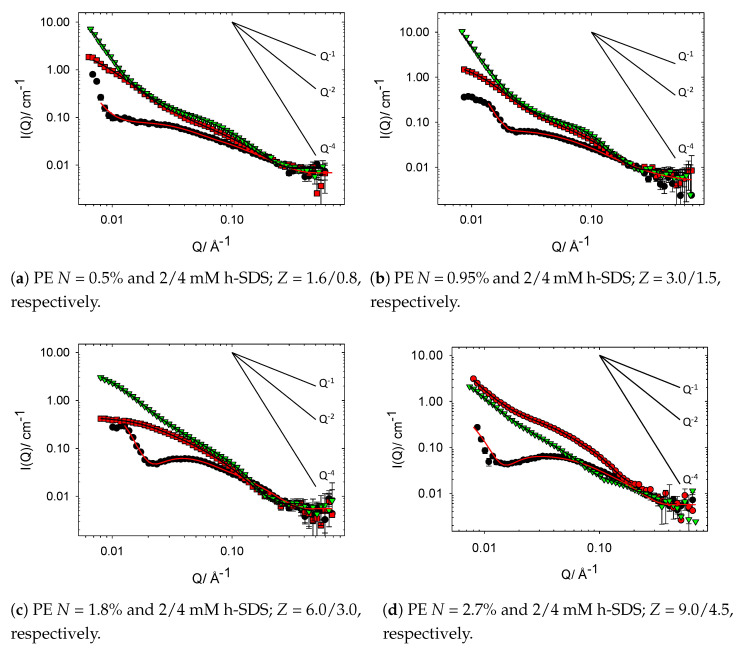
SANS profiles from binary mixtures containing 1 wt% cat-HEC polymers with different
degrees of charge modification and h-SDS (full contrast). Symbols: circles, no SDS; squares, 2 mM
SDS; down triangles, 4 mM SDS. The solid lines show Q^−1^, Q^−2^ and Q^−4^ behaviour.

**Figure 5 polymers-13-02800-f005:**
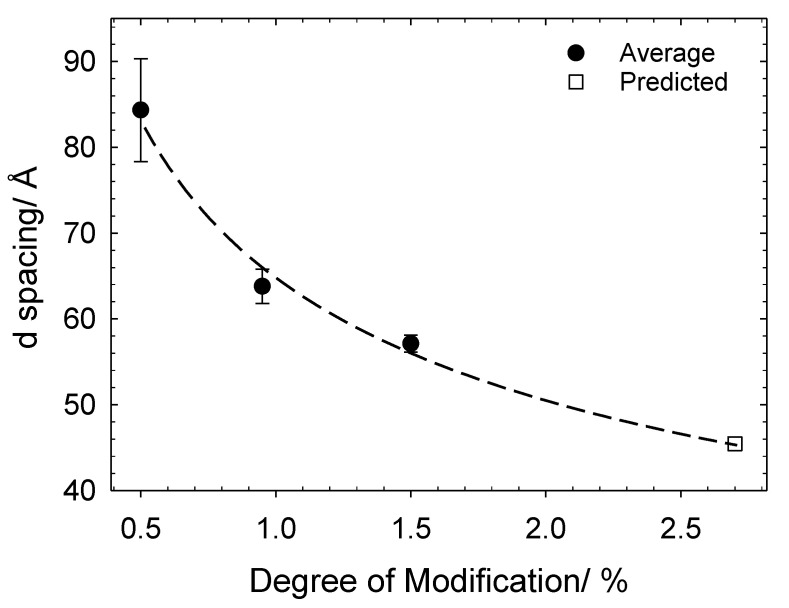
Change in d-spacing between surfactant complexes for the different cat-HEC PEs. Symbols: filled circles, averaged value; unfilled square, predicted value.

**Figure 6 polymers-13-02800-f006:**
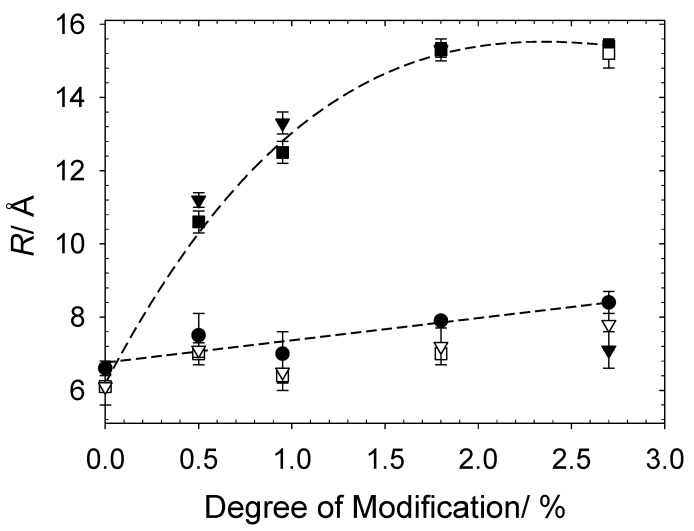
Fitted radius of 1 wt% cat-HEC polymers and cat-HEC/surfactant complexes (full and PE contrast). Symbols: circles, no SDS; filled squares, 2 mM h-SDS; unfilled squares, 2 mM d-SDS; filled down triangles, 4 mM h-SDS; unfilled down triangles, 4 mM d-SDS.

**Figure 7 polymers-13-02800-f007:**
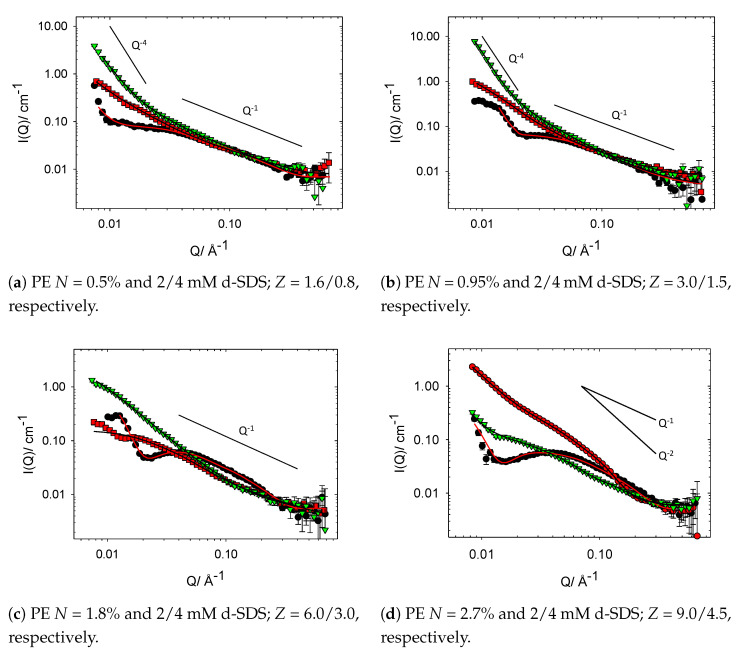
SANS profiles from binary mixtures containing 1 wt% cat-HEC polymers with different
degrees of charge modification and d-SDS (PE contrast). Symbols: circles, no SDS; squares, 2 mM
SDS; down triangles, 4 mM SDS.

**Table 1 polymers-13-02800-t001:** Fitted structural parameters of the family of cat-HEC polymers. *R* is radius, *L* is length, and *f* is fractal dimension.

Degree of Modification (%)	*R* (*Å)*	*L* (*Å*)	*f* (±0.2)
0	6.6	356	-
0.5	7.5	109	3.0
0.95	7.0	80	3.0
1.8	7.9	73	3.0
2.7	8.4	78	3.0

**Table 2 polymers-13-02800-t002:** Fitted structural parameters for binary mixtures containing 1 wt% unmodified polymer and h-SDS (full contrast) or d-SDS (PE contrast). *R* is radius and *L* is length.

Degree of Modification	Contrast	[SDS]	*R*	± 0.5	*L*
(%)		(mM)	(*Å*)		(*Å*)
0	Full	2	6.3		345
		4	6.2		278
	PE	2	6.1		300
		4	6.1		310

**Table 3 polymers-13-02800-t003:** Fitted structural parameters for binary mixtures containing 1 wt% cat-HEC polymers with different degrees of charge modification and h-SDS (full contrast) or d-SDS (PE contrast). *Z* is charge ratio, *R* is radius, and $N_{\mathrm{agg}}$ is surfactant aggregation number calculated using Equation (Equation 5). The fitted fractal dimension (*f*) is 3.0 ± 0.2 for all mixtures.

Degree of Modification (%)	Contrast	[SDS] (mM)	Z	*R* (*Å*)	$N_{\mathrm{agg}}$
0.5	Full	2	1.6	10.6	38 ± 7
		4	0.8	11.2	31 ± 6
	PE	2	1.6	7.0	-
		4	0.8	7.1	-
0.95	Full	2	3.0	12.5	39 ± 5
		4	1.5	13.3	40 ± 5
	PE	2	3.0	6.4	-
		4	1.5	6.5	-
1.8	Full	2	5.0	15.3	42 ± 9
		4	3.0	15.3	45 ± 7
	PE	2	5.0	7.0	-
		4	3.0	7.2	-
2.7	Full	2	9.0	15.4	-
		4	4.5	7.1	-
	PE	2	9.0	15.2	-
		4	4.5	7.8	-

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
