# Peer review of "Charge Modification as a Mechanism for Tunable Properties in Polymer–Surfactant Complexes"

_polymers, 2021, doi:10.3390/polym13162800_

Round 1
Reviewer 1 Report
Although the aim of this study was "improvement of understanding" into phenomena and dynamics of the sythems PEs and surfactant, it seems tha authors did not achieved clearconclusions that are supported undoubtly with experimental data. Also, a lot of references are present in results as well as in conclusion which caan reader lead to the conclusion that manuscript does not have novelty. Also, conlusion must be corrected as brings many unnecessery information. For example, authors repeat aim of the study and reasons for this investigation.
However, some data can be intersted to the certain number of readers and can be helpfull in future work in this field.
Reviewer 2 Report
Journal: Polymers
Manuscript number: 1317212
Title: Charge modification as a mechanism for tunable properties in polymer/surfactant complexes
Comments:
- The abstract and conclusion are both too long, and please concise it.
- The introduction should be updated and the research gap/objective are not clear here.
- The results should be further discussed.
- The graphs in this manuscript should be further improved.
- The English writing should be improved.
Reviewer 3 Report
The work by Hill et al. presents a very interesting work on polyelectrolyte-surfactant mixtures, with the main novelty arriving from the way in which the charge ratio is changeds. In this case, authors explore the change of the charge density as control parameter. Some minor aspects to solve:
- the work Curr. opin. Colloids Interface Sci. 2020, 48, 91-108.
- -authors should explain better the meaning of the catiónico substitution in the polymer.
- -what is the meaning of a small highly hydrated solución confirmation?
- -authors should be more concise in the description of the findings. There is so much information, and many times It is difficult to follow It
Round 2
Reviewer 1 Report
The manuscript is relativly improved and can be consider for publication after technical review.
Reviewer 3 Report
The manuscript is fully publishable now